# Lipschütz Ulcer and SARS-CoV-2: What We Currently Know?

**DOI:** 10.3390/diseases11030121

**Published:** 2023-09-12

**Authors:** Lucia Merlino, Agnese Immacolata Volpicelli, Mattia Dominoni, Marianna Francesca Pasquali, Giulia D’Ovidio, Barbara Gardella, Roberto Senatori

**Affiliations:** 1Department of Medical-Surgical Sciences and Biotechnologies, Sapeinza University of Rome, 00161 Rome, Italy; 2Department of Maternal Infantile and Urological Sciences, Sapeinza University of Rome, 00161 Rome, Italy; 3Department of Clinical, Surgical, Diagnostic and Paediatric Sciences, University of Pavia, 27100 Pavia, Italy; 4Department of Obstetrics and Gynecology, IRCCS Foundation Policlinico San Matteo, 27100 Pavia, Italy; 5Italian Society of Colposcopy and Cervicovaginal Pathology (SICPV), 00186 Rome, Italy

**Keywords:** Lipschütz ulcer, acute vulvar ulcer, non-sexually transmitted vulvar aphthosis, SARS-CoV-2, COVID-19

## Abstract

Background: In recent years, several interesting case reports have been published which describe the possible role of SARS-CoV-2 infection or vaccination in the etiopathogenesis of Lipschütz ulcer. Our aim is to analyze this association and provide a rapid algorithm that is of support to gynecologists and dermatologists both in the diagnosis and in setting up the therapy. To do so, in this paper, we describe an interesting case of acute vulvar ulcer triggered by SARS-CoV-2 infection and review the related literature. Methods: We conducted a literature review including papers published between October 2021 and April 2023, and we described the case of a patient referred to our clinic with Lipschütz ulcer and SARS-CoV-2 infection. Results: In almost all cases analyzed, a correlation with SARS-CoV-2 infection or vaccination was found; ulcers usually manifest after 2 to 4 weeks and are associated with flu-like symptoms. A concordance in review papers, as well as in our case report, was also found about the treatment, which is mainly symptomatic. Conclusions: Previous infection or vaccination for SARS-CoV-2 should be included as possible etiopathogenetic factors in the onset of Lipschütz ulcer.

## 1. Introduction

In clinical practice, the etiological identification of genital ulcers is still challenging, and sometimes the diagnosis and subsequent treatment can be difficult. The most common cause of vulvar ulcers is represented by infections, especially sexually transmitted ones (for example Herpes virus), although they can also be a manifestation of a wide range of diseases such as autoimmune diseases, malignancy, inflammatory processes, and trauma (including sexual abuse) [1]. 

Lipschütz ulcer (LU), also called “ulcus vulvae acutum” or “acute genital ulceration”, was first described in 1913 by Benjamin Lipschütz [2]. At first, it was thought to be a rare and likely underdiagnosed condition, but a recent study found that it may account for up to 30% of vulvar ulcerations [3]. The most affected are usually young women, with a mean age of 29.1 years, although some studies have reported a higher incidence in children and old women [4]. 

The clinical manifestation of Lipschütz vulvar ulcer is a sudden onset of painful ulcers, spread to any area of the vulva, apparently sine causa [5]. This kind of ulcer is typically 0.1 to 2.5 cm broad and deep, with a red border and a necrotic center covered by grey exudate or grey-black eschar, associated with labial edema, fever, and lymphadenopathy, and occasionally preceded by vague non-gynecological symptoms. ‘Kissing lesions’ frequently occur bilaterally. According to Moise et al., the etiology of Lipschütz ulcer is assumed to be a hypersensitive immune response to a bacterial or viral infection that causes the outbreak of immune complexes in the dermal arteries, leading to the development of microthrombi, which, in turn, provoke painful necrotizing ulcers [6]. After excluding all other potential causes of vaginal ulcerations, the diagnosis is primarily clinical and reached through exclusion; although some cases have indicated a connection with viral or bacterial infections, the etiology is unknown [4,7,8] These ulcers are self-limiting and spontaneously resolve within 2 to 6 weeks on average. The goal of treatment is supportive care, which includes pain control, prevention of secondary infections, reassuring patients, and offering guidance in advance [9].

The diagnostic criteria for Lipschütz ulcer are included below.

### 1.1. Major Criteria

Acute onset of at least one painful vulvar ulcer.Need to exclude infectious and non-infectious causes.

The most common infectious cause of ulcers is EBV, followed by Mycoplasma Pneumoniae, CMV, Toxoplasma gondii, HSV 1–2, Influenza virus, Mumps, Salmonella, or PVB19. Recently, primary infection with human immunodeficiency virus has also been recognized as one of the triggers of the onset of true ulcers. All these factors must be excluded through serological and culture tests.

The non-infectious causes are Bechet’s syndrome, Crohn’s disease, bullous diseases, cancer, and previous traumas. In Crohn’s disease, however, the ulcers are usually recurrent and located in the perianal region, and the patient also has gastroenterological signs. In Bechet’s syndrome there is usually a positive medical history for oral and genital aphthae, but also for uveitis and retinal vasculitis. It is advisable to ask for this information and, in suspicious cases (such as the presence of vulvar ulcers as the first manifestation), to require a rheumatological consulting to deepen the clinical picture [5].

### 1.2. Minor Criteria

Localized ulcer at the level of the vestibule or labia minora.Abstention from sexual intercourse in the last 3 months or patient who has never had sexual intercourse.Flu-like symptoms.Development of systemic infection at least 2–4 weeks preceding ulcer development [5].

Several trigger factors have been identified, such as Epstein–Barr Virus (EBV) or Cytomegalovirus (CMV). Primary EBV infection is the most frequently reported etiologic trigger in cases of young women who are not sexually active, and the ulcers brought on by the primary infection frequently appear before any other more typical symptom. Frequently, EBV genital ulcers are accompanied by secondary symptoms including malaise, headache, and low-grade fever; subsequently, the subject will develop more specific symptoms like tonsillitis, pharyngitis, or both, enlargement and discomfort of cervical lymph nodes, and moderate to high fever. Several hypotheses have been formulated regarding the etiology of primary EBV infection and Lipschütz ulcer; however, none of them have been demonstrated [10,11,12,13].

In addition, with the advent of the SARS-CoV-2 pandemic, several cases of Lipschütz ulcer development have been reported in conjunction with SARS-CoV-2 infection or vaccination. The COVID-19 vaccination has caused a number of cutaneous events, the most common of which include delayed large local reactions, local injection site reactions, urticarial eruptions, and morbilliform eruptions. It is interesting to note that several of these responses have been connected to COVID-19 infection.

With this paper, we conducted a review of the literature regarding the association between Lipschütz ulcer and SARS-CoV-2 infection and/or vaccination to understand whether this connection is present and, if so, how to manage these cases. Furthermore, we observed a case of a patient with Lipschütz ulcer which was triggered by SARS-CoV-2 infection in our clinic and we promptly reported it here as a Case Report.

## 2. Materials and Methods

The medical literature regarding the association between Lipschütz ulcer and SARS-CoV-2 infection or vaccination was retrieved consulting PubMed, Cochrane Database of Systematic Reviews, EMBASE, and Web of Science. The search strategy included a combination of the following keywords: “Lipschütz ulcer”, “acute vulvar ulcer”, “non-sexually transmitted vulvar aphthosis”, AND “COVID-19” or “SARS-CoV-2”. We retrieved all articles published between October 2021 and April 2023, which were all case studies; no prospective or retrospective clinical trials were found. Only articles written in English were considered. To include the data from the literature in the review, two authors (L.M. and R.S.) independently assessed the references. The articles were then selected based on the abstract and, subsequently, on the full text content; 14 of them were considered eligible and, thus, analyzed.

From the 14 articles, 23 clinical cases were obtained. The symptomatology of each case was analyzed; specifically, the time elapsed between the infection or vaccination and the development of the ulcer (48–72 h), the type of vaccine, and the time in which the ulcer was resolved (2–3 weeks). 

Studies were considered qualified if they met either of the following criteria: (I) The development of vulvar ulcer with SARS-CoV-2 infection, or (II) the role of the anti-SARS-CoV-2 vaccination in vulvar disease. The exclusion criteria were: (I) conference abstracts, editorials, and preprints manuscript; (II) multimedia; or (III) experimental papers with results based on animal experimentation.

Two researchers (L.M and R.S.) independently evaluated the risk of bias for each selected study in accordance with the Cochrane Handbook for Systematic Reviews of Interventions [14,15] to assure validity and prevent any selection, performance, detection, attrition, and reporting bias. Finally, each researcher independently analyzed and retrieved data. 

Regarding our case report, the patient in this manuscript gave written informed consent to publication of her case details.

## 3. Results

The present paper is divided into two different sections: the first part reports the clinical presentation of our case, while the second part highlights the results of the literature review. 

The analysis of previous data has demonstrated that the literature currently available about the development of Lipschütz ulcer and SARS-CoV-2 vaccination and infection is essentially based on few clinical reports of cases, reviews, or clinical retrospective or prospective papers. To our knowledge, this is the first review on this clinical disease related to SARS-CoV-2 infection. 

### 3.1. Case Report

We describe the case of a 15-year-old student that attended our clinic in February 2023, accompanied by her mother, complaining of localized vulvar pain and describing the presence of suspicious lesions in the vulvar area. The patient presented with flu-like symptoms, which she experienced for at least a week, but had never investigated or treated. 

Through a careful vulvoscopic examination, we observed multiple lesions compatible with vulvar ulcers that, thanks to the magnification of the colposcope, appeared deep with a necrotic aspect, with a healthy fundus, superimposed on a hyperemic border. They were distributed at the level of the inner face of the labia minora bilaterally and at the vestibule: one right, around 5–6 mm, and two left, around 15–20 mm of maximum diameter, described as extremely painful. 

Figure 1 describes the clinical picture at the baseline.

At the general physical examination there were no notable clinical signs other than a flu-like state. The girl had never had sexual intercourse. She had a silent personal and family history, with no cases of autoimmune disease. She had correctly performed all the vaccinations for SARS-CoV-2 and Human Papilloma Virus (HPV). She did not take any medications and she denied allergies. Psychosocial history showed nothing remarkable. 

We performed a complete blood count, which showed no alterations. CRP (C Reactive Protein) level was elevated (20 mg/L). Serological investigations for HSV 1 and 2, EBV, CMV, VZV, HBV, HCV, HIV, and cultures for Mycoplasma pneumoniae, Trichomonas vaginalis, and Toxoplasma gondii were all negative. Due to her flu-like symptoms, we also investigated the presence of SARS-CoV-2 with a nasopharyngeal swab, which turned out positive.

Considering the diagnostic criteria for Lipschütz ulcer [5], and having excluded the other causes, we made the diagnosis. Considering the results of the test, we deduced that, in this case, the trigger was SARS-CoV-2 infection.

The patient was treated with 4 mg of methylprednisolone and ibuprofen orally for a total of 6 days, and locally with eosin and rice starch.

She tested negative for SARS-CoV-2 after 5 days and returned to our clinic after 10 days, showing an improvement in the clinical picture (Figure 2). She achieved complete remission after 15 days.

### 3.2. Literature Review 

The main characteristics of the 14 papers analyzed are summarized in Table 1, resulting from the research carried out with the indications exposed in Materials and Methods. Out of a total of 23 cases of Lipschütz ulcer, 5 are related to SARS-CoV-2 infection, while 18 are related to COVID-19 vaccination. 

Alberelli et al. [16], Christl et al. [17], Jacyntho et al. [18], Morais et al. [19], and Schmitt et al. [20] presented case reports of Lipschütz ulcer linked to SARS-CoV-2 infection. The patients’ ages ranged from 10 to 19 years old, with the exception of a 35-year-old in the work of Jacintho et al. [18] Two of them have had previous intercourses, but the serological and microbiological tests, when performed, resulted negative. Women usually presented with vulvar pain and flu-like symptoms, and, in this context, developed Lipschütz ulcer, as happened to the patient in our case report. In one case, urinary symptoms were associated. The proposed treatment, mostly based on the control of vulvar pain, consisted of oral NSAIDs, usually in association with topical lidocaine (Christl et al. [17] and Morais et al. [19]). In one case (Jacintho et al. [18]), due to severe and progressive vulvar pain, oral corticosteroids were needed. 

In all cases the ulcers resolved within 1 to 4 weeks. 

Concerning the cases of Lipschütz ulcer linked to COVID-19 vaccination, symptoms arose on average 2 days after the administration of the vaccine. In all cases, except for Sartor et al. [21], the type of vaccine was specified; seven cases (50%) were linked to Pfizer, one to Moderna, and two to AstraZeneca. 

The women were between 12 and 33 years old, and only one of them had previous sexual intercourse. However, serological and microbiological examinations were often performed that tested negative.

The main clinical presentations were flu-like symptoms associated with vulvar pain after the onset of the ulcers. In three cases (Schmitt et al. [20], Scott et al. [22], and Wojcicki et al. [9]) more severe systemic symptoms, like myalgia, fatigue, headache, and nausea occurred. Only one patient had a high fever, >38 °C, treated with oral paracetamol. 

Frederiks et al. [23] reported a particular case of Lipschütz ulcer where the patient, in addition to vulvar pain, had constipation and was in fact treated for both, with topical steroids for the vulvar pain and with an osmotic laxative for the constipation.

Hsu et al. [24] described the clinical case of a 12-year-old patient who presented with painful ulcerations associated with dysuria and hematuria after the vaccine. She was given oral nitrofurantoin, fluconazole, and phenazopyridine with the remission of her urinary symptoms and, subsequently, diagnosed and treated for vulvar ulcers.

The proposed treatments were generally based on oral NSAIDs for pain relief, which were administered to all patients, and topical anesthetics, such as lidocaine 5%. In seven cases, topical corticosteroids were associated, while oral corticosteroids were needed in four patients to counteract invalidating vulvar symptoms. 

Sartor et al. had the most extensive case history, with eight cases of Lipschütz ulcer linked to vaccination against SARS-CoV-2. The patients’ ages ranged between 12 and 17 years old, the ulcers occurred 2 weeks after vaccination, and the proposed treatment was based on topical lidocaine and NSAIDs for pain symptoms; topical corticosteroid was also administered in cases of severe pain.

Consistent with the literature, we can elaborate the following conclusions:The ulcers predominantly affect adolescents; in fact, there are only two cases of older patients, aged 29 and 35 years old.The most common clinical manifestations are represented by flu-like symptoms and vulvar pain.If the ulcer is linked to the vaccine, it usually manifests within 48–72 h following the vaccination.As per protocol, all other possible infectious causes were excluded in each patient by means of serological and culture investigations.The resolution occurs spontaneously within 2–3 weeks.Since the treatment is purely symptomatic, in most cases the therapy consisted of a local anesthetic like lidocaine for topical use and oral NSAIDs to help control pain, sitz baths for greater intimate hygiene, and, in some cases, the use of topical corticosteroid. The use of the latter was observed in more complex cases and, above all, in cases in which the patient was particularly suffering for which there was a need for more aggressive treatment.
diseases-11-00121-t001_Table 1Table 1Cases in the international scientific literature that correlate the development of acute ulcer at the level of the vulva with vaccination against SARS-CoV-2 or SARS-CoV-2 infection.Authors and YearMedian AgePrior Sexual ActivitySymptomsOral UlcerInfectious Cause (HSV, CMV, EBV, HCV, HBV *Mycoplasma pneumoniae*, *Toxoplasma gondii*)Previous Vaccination for COVID-19Previous or Concomitant SARS-CoV-2 Treatment RemissionAlberelli et al., 2021 [16]10NoFever, vulvar painNoNegative. Toxoplasma not performedUndeclaredYesGentamicin cream and oral amoxicillin, ibuprofen as needed for pain2 weeksChristl et al., 2021 [17]19YesFlu-like syndrome, vulvar pain YesNot performedUndeclaredYesLidocaine for topical use, ibuprofen orally for pain control for 2 weeks, dexamethasone after for 1 week3 weeksDrucker et al., 2022 [25]14UndeclaredVulvar pain after the vaccineNoHSV, CMV, EBV negative. Others not performedYes (Pfizer)NegativeLidocaine for topical use and sitz baths10 daysFrederiks et al., 2022 [23]12NoConstipation and vulvar pain 3 days after the vaccineNoNegative. HBV, HCV not performedYes (Pfizer)Not performedOral analgesia, laxatives, medium strength topical corticosteroid ointment10 daysGonzález-Romero et al., 2022 [26]24NoFlu-like symptoms and vulvar pain after the vaccineNoHSV, HBV, HCV, Mycoplasma negative. Toxoplasma, CMV, EBV not performed Yes (AstraZeneca)Not performedPrednisone (30 mg) and oral analgesics3 weeksHsu et al., 2022 pz1 [24]12NoDysuria, hematuria, flu-like symptoms 2 days after the vaccineNoHSV and EBV negative. Others not performedYes (Pfizer)NegativeTopical clobetasol 0.05%, topical lidocaine 2%, as needed oral acetaminophen and ibuprofen, and sitz baths2 weeksHsu et al., 2022 pz2 [24]14NoVulvar pain caused by ulcers that arose 3 days after the vaccineNoHSV and EBV negative. Others not performedYes (Pfizer)NegativeTopical clobetasol 0.05%, topical lidocaine 2%, as needed oral acetaminophen and ibuprofen, and sitz baths4 weeksHsu et al., 2022 pz3 [24]29NoFlu-like symptoms 23 h after the vaccineYesHSV, CMV, EBV negative. Others not performedYes (Moderna)NegativeTopical clobetasol 0.005%2–4 weeksLucero Sangster-Carrasco et al., 2023 [27]33YesVulvar pain and malaise 3 days after the vaccine YesHSV, CMV, EBV, Toxoplasma negative. Mycoplasma, HCV, HBV not performedYes (AstraZeneca)UndeclaredOral antibiotics and oral NSAIDs 2 weeksJacyntho et al., 2022 [18]35NoProgressive severe vulvar painNoNegativeUndeclaredYesOral NSAIDs and oral corticosteroid 1 weekMorais et al., 2022 [19]17YesFlu-like symptoms, itching, dysuriaNoHSV, mycoplasma CMV, EBV negative. Toxoplasma, HCV, HBV not performedUndeclaredYesOral NSAIDs and lidocaine for topical use2 weeksPopatia et al., 2022 [28]12NoFever 1 day after the vaccine NoHSV negative. Others not performedYes (Pfizer)NegativeLidocaine 2% jelly, triamcinolone 0.1%, acetaminophen and ibuprofen for pain10 daysSartor et al., 2023 pz1 [21]17UndeclaredVulvar pain 2 days after the vaccine UndeclaredHSV negative. Others not performedYesUndeclaredMethylprednisolone taper, amoxicillin, lidocaine for topical use, NSAIDs, sitz baths 20 daysSartor et al., 2023 pz2 [21]14UndeclaredVulvar pain 2 days the vaccine UndeclaredHSV negative. Others not performedYesUndeclaredLidocaine for topical use, NSAIDs25 daysSartor et al., 2023 pz3 [21]13UndeclaredVulvar pain 2 days the vaccine UndeclaredHSV negative. Others not performedYesUndeclaredOral prednisone, cephalexin, lidocaine for topical use, NSAIDs, sitz baths14 daysSartor 2023 pz4 [21]12UndeclaredVulvar pain 2 days the vaccine UndeclaredCMV, EBV negative. Others not performedYesUndeclaredOral steroids21 daysSartor 2023 pz5 [21]14UndeclaredVulvar pain 2 days after the vaccine UndeclaredNot performedYesUndeclaredUndeclared21 daysSartor 2023 pz6 [21]15UndeclaredVulvar pain 2 days after the vaccineUndeclaredEBV negative. Others not performedYesUndeclaredUndeclared21 daysSartor 2023 pz7 [21]16UndeclaredVulvar pain 2 days after the vaccine UndeclaredNot performedYesUndeclaredZinc oxide, NSAID, lidocaine for topical use10 daysSartor 2023 pz8 [21]16UndeclaredVulvar pain 2 days after the vaccineUndeclaredNot performedYesUndeclaredZinc oxide, NSAID, lidocaine for topical use10 daysSchmitt 2023 [20]15NoNausea, headache, fever of 38.3 °C, vulvar painNoHSV, CMV, EBV negative. Others not performedNoYesAlternating paracetamol or ibuprofen and cold compresses14 daysScott 2022 [22]15NoVaginal pain and sores, 3 days after the vaccine NoNot performedYes (Pfizer)Not performedSitz baths, oral pain relievers, topical anesthetics, topical steroids1 weekWojcicki 2022 [9]16NoVaginal pain, myalgia, and fatigue 2 days after the vaccineYesHSV, CMV, EBV negative. Others not performedYes (Pfizer)NegativeTopical lidocaine, ibuprofen and paracetamol for pain control, topical Clobetasol, norethindrone (5 mg) for menstrual suppression while ulcers heal14 days


## 4. Discussion

Lipschütz vulvar ulcer is a self-limiting painful ulcer with an acute onset that typically affects sexually inactive adolescents [29]. However, data from the recent literature have shown an association with bacterial or viral infections in some cases, laying the groundwork for a possible correlation with sexually transmitted diseases. The main infections reported are EBV, Mycoplasma Pneumoniae, CMV, Toxoplasma gondii, Herpes Simplex Virus 1–2 (HSV 1–2), Influenza virus, Mumps virus, Salmonella, or PVB19. From our review, we can observe how both SARS-CoV-2 infection and previous COVID-19 vaccination can be recognized as trigger factors [5]. For this reason, the etiopathogenesis it is not entirely clear. 

The factors listed above, considering what is present in the literature, can determine the production of immune complexes in the acute phase of infection, leading to the formation of a type III hypersensitivity reaction: the immune complexes react with the complementary system, causing the triggering of an inflammatory response which leads to microthrombosis and tissue necrosis [30].

This hypothesis is also supported by the histological examination of Lipschütz ulcer biopsies described in the case report by Sárdy et al., which showed how the histology of the ulcer, in this case associated with EBV infection, is consistent with a localized immune complex vasculitis [8].

The ideal strategy would be to perform biopsies on several samples, with different clinical histories and causes, to be able to confirm these data. However, biopsy investigation is almost never performed nowadays because it does not lead to a change in the therapeutic plan.

Vaccination against SARS-CoV-2, as well as SARS-CoV-2 infection, also determines a strong inflammatory response that would be compatible with the pathogenesis proposed above for the development of vulvar ulcers. Cutaneous manifestations that emerged during SARS-CoV-2 infection were observed and analyzed, and, from a histological point of view, they resulted in a pauci-inflammatory thrombogenic vasculopathy, similar to the histological manifestation of the vulvar ulcer associated in the previous case with EBV [31].

In our review, we demonstrate how infection or previous vaccination for SARS-CoV-2 can be included as triggers for the onset of Lipschütz ulcer (Figure 3). Regardless, it is always of primary importance for the diagnosis of Lipschütz ulcer to start from an extensive clinical history, excluding autoimmune diseases, and physical examination, followed by serological screening and culture tests for the main risk factors [6]. However, if all those test results are negative, considering the frequent association with flu-like symptoms, it is advisable to require a nasopharyngeal swab for SARS-CoV-2. At the same time, it is in the interest of the clinician to ask how long since the ulcers appeared, as well as when the infection arose or when the vaccination for SARS-CoV-2 was administered. If either the infection or the vaccine was within 2–4 weeks preceding the development of the ulcer, after excluding all other potential causes, then it is possible to recognize one of these two elements as a trigger.

In all selected studies of our review there has always been concordance about the treatment, which is mainly symptomatic. Since the lesions heal spontaneously, the therapy is therefore mostly aimed at promoting pain relief. Usually, a local support medication with topical lidocaine 5% and oral NSAIDs for flu-like symptoms are sufficient, but in cases of intense pain not adequately supported by local control, a systemic approach can be attempted. Generally, oral corticosteroids are administered with a good control of symptoms, although they do not accelerate the remission of the ulcers. In our case report, in accordance with the literature, we treated the patient with local lidocaine 5%, corticosteroid, and oral NSAIDs with complete control of the symptoms. 

## 5. Conclusions

Lipschütz ulcer is a very painful clinical pathology and can be experienced with great anxiety by the patient. It is advisable, in case of suspicious lesions, to refer to the diagnostic criteria, to exclude an infectious cause by running serological and culture investigations, to rule out the presence of auto-immune diseases, to require a nasopharyngeal swab to assess SARS-CoV-2 infection, and to investigate whether the patient has previously been vaccinated for SARS-CoV-2. It is important to reassure the patient, explaining that the ulcers are self-limiting and resolve spontaneously within 1 to 4 weeks, they usually are not related to sexually transmitted diseases, and we can provide an effective supportive therapy.

This study has some limitations: Due to the absence of systematic reviews and experimental clinical trials, the literature review is based on few individually documented case reports. This implies that a formal assessment of bias was not possible. On the other hand, in this paper, the correlation between SARS-CoV-2 infection or COVID-19 vaccination and the onset of Lipschütz ulcer was analyzed for the first time, and we pointed out the most used treatments of this disease. Given the scant literature about this pathology, more studies are surely needed to implement the knowledge regarding the etiopathogenesis, diagnosis, and treatment of this disease. It would be interesting to develop and evaluate the pathology on a prospective study to obtain more accurate results.

## Figures and Tables

**Figure 1 diseases-11-00121-f001:**
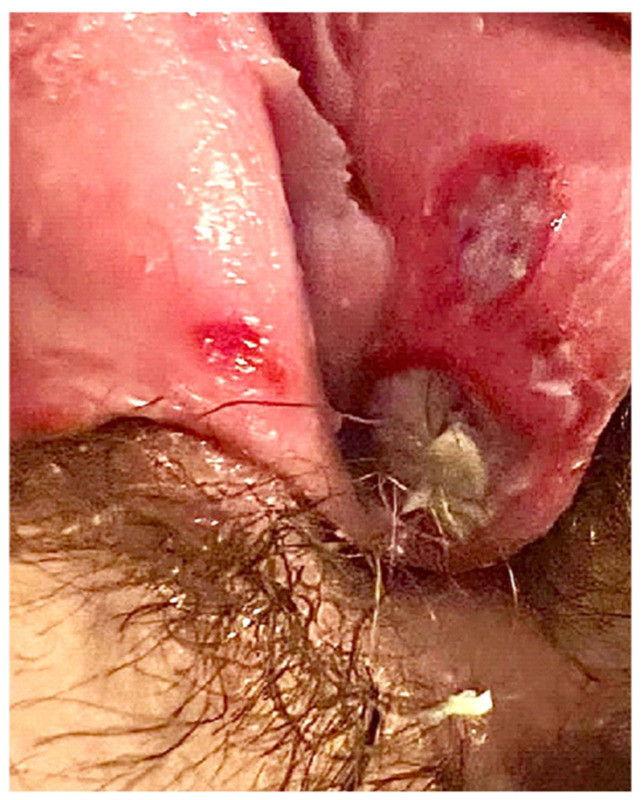
Ulceration of the inner face of the labia minora bilaterally and at the vestibule, with hyperemic and purplish margins.

**Figure 2 diseases-11-00121-f002:**
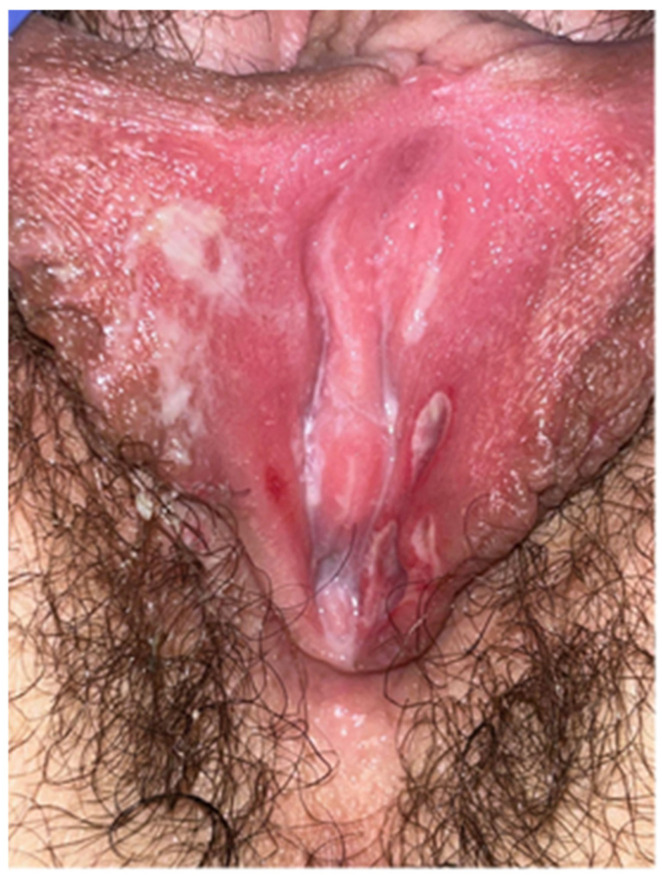
Improvement of the clinical picture after 10 days.

**Figure 3 diseases-11-00121-f003:**
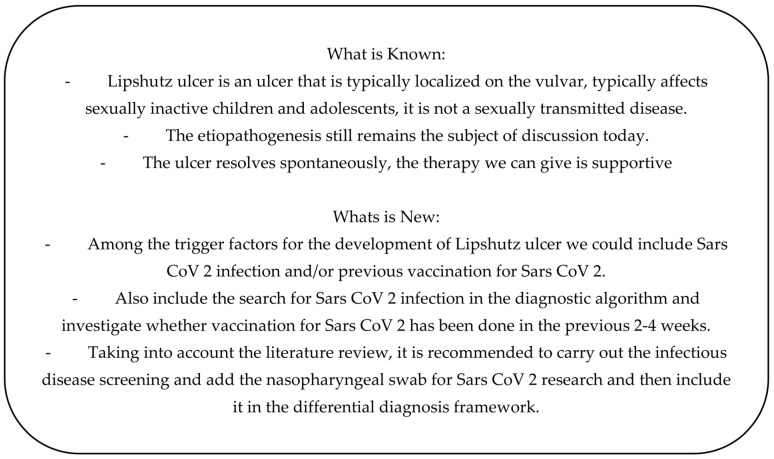
Summary of the most significant points between Lipschütz ulcer and SARS-CoV-2.

## Data Availability

Not applicable.

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
