# Peer review of "Lipschütz Ulcer and SARS-CoV-2: What We Currently Know?"

_diseases, 2023, doi:10.3390/diseases11030121_

Round 1

Reviewer 1 Report

Very unique study to be considered for publication.

Reviewer 2 Report

The authors tried to find a corretion between vaccination or SARS-Cov-2 infection and the existence of ulcer.

Please described the MM section better as the duration of SARS-Cov-2 infection, the kind of vaccination, doses of vaccination.

Please explain why vaccination promotes the ulcer development, is there any correlation?

Minor typos and laguage editing is needed

Reviewer 3 Report

Interesting,no critical remarks 

Reviewer 4 Report

Dear author’s 

I was pleased to review your article and I have the following comments:

The subject is debatable. We cannot conclude that this pathology is secondary to covid infection or vaccination.

There are multiple repetitive sentences belong the article sections like “ Lipschütz vulvar ulcer was first described in 1913” etc. Please remove the repetitive information. 
It will be interesting to add pictures with the colposcopic examination.

 It is mandatory to draw a prospective study in order to obtaine accurate results.

Minor English edits.

Round 2

Reviewer 4 Report

Thank you for your revised version.